# Heterogeneous Causal Relationships between Spot and Futures Oil Prices: Evidence from Quantile Causality Analysis

**Xianfang Su** [1,*] **, Huiming Zhu** [2] **and Xinxia Yang** [3]

[1] School of Big Data Application and Economics, Guizhou University of Finance and Economics, Guiyang 550004, China
[2] College of Business Administration, Hunan University, Changsha 410082, China; zhuhuiming1998@126.com
[3] School of Mathematics and Statistics, Guizhou University of Finance and Economics, Guiyang 550004, China; yangxinxia@126.com
[*] Correspondence: xfsu@hnu.edu.cn

**Abstract:** The causal relationships between spot and futures crude oil prices have attracted the attention of many researchers in the past several decades. Most of the studies, however, do not distinguish among the various oil market situations in analyses of linear and nonlinear causalities. In light of the fact that a booming or depressing oil market produces heterogeneous investment behaviors, this study applied a quantile causality framework to capture different causalities across various quantile levels and found that the causal relationships between crude oil spot and futures prices significantly derive from tail quantile intervals and appear as heterogeneous effects. Before the Iraq War, crude oil spot and futures prices were mutually Granger-caused at lower quantile levels, and only futures prices led spot prices at upper quantile levels. Since the war, a clear bidirectional causality has existed at the upper quantile levels, but only in lower quantile levels have futures prices led spot prices. These results provide useful information to investors using crude spot or futures prices to hedge or manage downside or upside risks in their portfolios.

**Keywords:** crude oil market; futures price; heterogeneous relationship; quantile causality test

## 1. Introduction

The causal relationships between crude oil spot and futures prices have long been under scrutiny by investors, policymakers, and researchers for an understanding of the price discovery mechanism in crude oil spot and futures markets, as well as an examination of the efficiency of the oil markets. However, the available empirical evidence on the causal relationships between oil spot and futures prices is mixed. Many early researchers suggested that futures prices responded to new information more quickly than did spot prices because of the lower transaction costs and of the flexibility of short selling. Hence, the crude oil futures markets would dominate the spot markets in price discovery [1–4]. In contrast, Quan [5] concluded that futures prices did not play a very important role in price discovery. Moosa [6] pointed out that changes in spot prices triggered actions from the three kinds of market participants and that these actions subsequently changed futures prices. These inconsistent empirical results indicate that the potential causality patterns between oil spot and futures prices may be complex and that neither market leads or lags behind the other consistently. Recently, many researchers have revealed nonlinear causal relationships between crude oil spot and futures prices [7–13]. These nonlinearities were attributed to nonlinear transaction cost functions, the role of noise traders, market microstructure effects, the diversity in agents' beliefs [14], herd behavior [15], oligopolistically and monopolistically competitive markets [16], and the heterogeneity

of investors' objectives [17]. However, these previous studies have overlooked the possibility that the causal relationships between crude oil spot and futures prices may differ according to different market conditions. In fact, it is important to consider the effects of market conditions on the causal relationships between spot and futures prices. This consideration can be illustrated by the fact that, although abundant empirical studies have been conducted on the causal relationships between crude oil spot and futures prices, there are still many deviations when compared with real decisions, especially for markets in extreme states.

This study analyzed the heterogeneous causal relationships between crude oil spot and futures prices under different market conditions. The underlying reason why the causal relationships may differ according to market conditions is the differential reactions produced by a booming or depressing market condition and their influences on real decision-making. In the crude oil spot–futures markets, if the relationships between the spot and futures prices do not approach equilibrium, then effects such as arbitrage and speculation will produce equilibrium [4], i.e., influence the causal relationships between crude oil spot and futures prices. According to the prospect theory proposed by Kahneman and Tversky [18], an investor feels more pain from a loss than happiness from a gain equivalent to the loss. In fact, arbitragers and speculators have different attitudes toward gains and losses, so their hedging and extreme risk management decisions differ according to market conditions. Therefore, the causality relationships between spot and futures crude oil prices may present heterogeneous effects across the quantiles of crude oil spot and futures price distributions. This possibility inspired us to apply a quantile causality analytical framework to investigate the extent to which heterogeneity exists in the causality relationships.

To capture the heterogeneous causal relationships across different market conditions, this research applied a two-stage approach to characterize the causal relationships between crude oil spot and futures prices. In the first step, this research used a vector error correction model (VECM) to filter the long-run co-movements in the level of prices. This is a reasonable setting for several reasons. On the one hand, it is quite well-known that the spot and futures prices are cointegrated, so it is necessary to control the long-run co-movement effects because they affect the specifications of the model used for causality testing. If the long-run co-movement is not modeled, the evidence may vary significantly toward detecting linear and nonlinear causalities between the predictor variables [10]. On the other hand, the VECM has been widely used in empirical studies to filter price series from long-run co-movements [19–24]. In the second step, after having filtered the price series with the VECM model, the series of residuals were examined with the quantile causality test proposed by Chuang et al. [25]. In contrast to a traditional linear estimation conditional on the mean distributions of dependent variables, this test allows us to explore the causality relationships conditional on the quantiles in the distributions of crude oil spot and futures prices. Because the quantile causality test considers different locations and scales of the conditional distribution, it may provide a complete description of the true causal relationships and can accommodate heterogeneous effects to allow us to capture the different responses of one market to another market at various quantile levels. Furthermore, the quantile levels for the conditional distribution of economic variables can indicate the states of the market, and the quantile causality test provides an efficient tool for handling this nonlinearity in the causal relationships.

The two-stage causal analysis framework is one of the contributions of this paper to the existing literature. The previous models that took the market's effects on the crude oil spot–futures relationships into account include mainly the structural break [26–28], multivariate threshold regression [12,29], and quantile cointegrating models [30]. However, although the structural breaks and threshold methods can be used to explore the crude oil spot–futures relationships among special market states, how the causal relationships between spot and futures prices change across the quantiles of price distribution is little known. The quantile cointegration method allows for quantile-varying in a cointegrating vector and can offer much information for an understanding of the nonlinear relationships in various market conditions. However, the quantile cointegrating regressions can estimate the long-run relations

between spot and futures oil prices conditional on market innovations but cannot explain the short-term dynamic relationships under various market conditions. Therefore, this study employed a VECM together with a quantile causality test to analyze the causal relationships between spot and futures prices on the basis of the performances of the crude oil markets. The two-stage model provides a flexible procedure and expands the available information set by considering long-run co-movements and by capturing various market states.

In the empirical analysis, apart from the heterogeneous causal relationships between crude oil spot and futures prices across extreme booming and depressing market conditions, this study also tested the causality relationships between crude oil spot and futures prices before and after the onset of the Iraq War (After the Iraq War broke out, the average level and volatility of crude oil prices showed an accelerated rise. Moreover, the Iraq War has been detected as a structural change point [31]). The key findings of this paper are as follows.

First, this study provides new evidence that the causal relationships between crude oil spot and futures prices present significant heterogeneous behaviors under different market conditions: The causal relationships between crude oil spot and futures prices over only the lower and upper quantile intervals but not the middle quantile intervals. The causal relationships between crude oil spot and futures prices are weaker when oil prices fluctuate at nearly their medians and are highly likely to be present when one market shows very good or very poor performance. This empirical result expands on and refines the findings of Huang et al. [29], as well as those of Wang and Wu [12], who suggested that at least one causal relationship between spot and futures prices exists only when the price differentials are larger than a threshold value. The results of this research were able to capture the causal relationships between crude oil spot and futures prices at specified quantile levels. Moreover, as compared to the conclusion of Lee and Zeng [30], this study found that crude oil spot and futures prices were influenced by quantiles and futures contracts, but they had found that spot oil prices indeed caused futures oil prices. This finding was not apparent in this study. The difference in the sample interval may be the underlying reason for the inconsistency.

Second, this study has revealed the nonlinear causality between crude oil spot and futures prices from the perspectives of different market states. The direction and significance of causal relationships between crude oil spot and futures prices changed with different market conditions. This result is in accordance with Bekiros and Diks [10], who suggested that if nonlinear effects were accounted for, neither market led or lagged behind the other consistently. Chang and Lee [31] used a wavelet coherency method and studied the time-varying causal relationships between crude oil spot and futures prices. Their results show a significant dynamic causality between variables in the time–frequency domain. Here, this research considers the nonlinear drivers, which are different from the focus of previous studies, and obtains a consistent conclusion.

Third, this study observed that the causal relationships between crude oil spot and futures prices before and after the Iraq War were distinctly different. After the Iraq War, the international crude oil markets faced more volatility, which stimulated different speculative and arbitrage behaviors to influence the causal relationships. This result verifies the finding of Fan and Xu [32], who pointed out that the Iraq War had been detected as a structural change point.

The remainder of the paper is organized as follows. Section 2 discusses the methodology. Section 3 reports the data and descriptive statistics. Section 4 provides empirical results, and Section 5 presents the conclusions.

## 2. Methodology

### 2.1. Filtering the Long-Run Co-Movement

The concept of cointegration has addressed the characteristic co-movement of commodity price series. Cointegration is the idea that, even though individual price series are nonstationary, a linear combination of price series may be stationary. The VECM is a vector autoregression (VAR) model with

cointegration restrictions. Moreover, the VECM includes an examination of the dynamic co-movements among variables and the adjustment process toward long-term equilibrium. To filter the crude oil spot and futures prices from long-run co-movements, this study applied a VECM to specify the conditional mean.

Let $p_t = (p_{S,t}, p_{F,t})'$ be a $(2 \times 1)$ vector of price series, $p_{S,t}$ denote the crude oil spot price at time $t$, and $p_{F,t}$ denote the crude oil futures price at time $t$. If the elements of $p_t$ are $I(1)$ variables, then the conventional VECM is represented as

$$\Delta p_t = c + \Pi p_{t-1} + \sum_{i=1}^{k} \Gamma_i \Delta p_{t-i} + \varepsilon_t, \tag{1}$$

where $\Delta$ is a first-difference operator such that $\Delta p_t = p_t - p_{t-1}$ denotes the change in the price series from time $t-1$ to time $t$ and $\Pi$ is a matrix containing long-run equilibrium information. If $rank(\Pi) = r < m$ and if $m$ is the number of parameters in the estimation model, then $\Pi$ can be written as $\Pi = \alpha\beta^T$, and the VECM form can be expressed again as

$$\Delta p_t = c + \alpha\beta^T p_{t-1} + \sum_{i=1}^{k} \Gamma_i \Delta p_{t-i} + \varepsilon_t \tag{2}$$

where $\beta^T p_{t-1}$ denotes the cointegration relation, $\alpha$ is the speed of adjustment at which the price series return to the long-run equilibrium, $\Gamma$ measures the short-run dynamic relationship between the elements of $p_t$, and $\varepsilon_t$ is a residue term that captures the filtered information. A quasi-maximum likelihood method can be used to estimate the parameters of VECM under the assumption of homoscedastic errors. Bauwens et al. [33] proposed that the estimation results were still consistent under the presence of heteroscedasticity errors.

### 2.2. Testing Non-Causality in Quantiles

After having filtered the price series with the VECM model, the series of VECM residuals were examined with the quantile causality test. In the following, this study will assume that $x_t$ and $y_t$ is the VECM residuals. For a comprehensive understanding of causal relationship between $x_t$ and $y_t$, the causality test in quantile proposed by Chuang et al. [29] is represented as

$$Q_{y_t}(\tau | \mathcal{Y}_{t-1}, \mathcal{X}_{t-1}) = Q_{y_t}(\tau | \mathcal{Y}_{t-1}), \quad \forall \tau \in [a, b] \subset (0, 1), \tag{3}$$

where $Q_{y_t}(\tau | \mathcal{M})$ denotes the $\tau$ quantile of the conditional distribution of $F_{y_t}(\cdot | \mathcal{M})$ and $\mathcal{M}$ denotes the information setup to time $t$, which could include $(\mathcal{Y}_{t-1}, \mathcal{X}_{t-1})$ or $\mathcal{Y}_{t-1}$. If Equation (3) holds, then we can say that $x$ does not Granger-cause $y$ over the quantile interval $[a, b]$. Contrarily, when Equation (3) fails to hold, the variable $x$ is said to Granger-cause variable $y$ over the quantile interval.

Granger non-causality in quantiles can be tested by the usual quantile regression method proposed by Koenker and Bassett [34], specified, for the $\tau$ conditional quantile, as

$$Q_{y_t}(\tau | \mathcal{M}) = \omega(\tau) + \sum_{i=1}^{p} \alpha(\tau)_i y_{t-i} + \sum_{j=1}^{q} \gamma(\tau)_j x_{t-j} \tag{4}$$

Therefore, if the parameter vector $\gamma(\tau) = (\gamma_1(\tau), \gamma_2(\tau), \ldots, \gamma_q(\tau))^T$ is equal to zero, then we can say that $x_t$ does not Granger-cause $y_t$ at the $\tau$ quantile level. Koenker and Bassett [34] proposed a sup-Wald statistic to test the null hypothesis of non-causality over a quantile interval.

For a specific $\tau$ quantile, the Wald statistic of $\gamma(\tau) = 0$ can be written as

$$W_T(\tau) = T \frac{\hat{\gamma}(\tau)' \hat{\Sigma}(\tau)^{-1} \hat{\gamma}(\tau)}{\tau(1 - \tau)} \tag{5}$$

where $\hat{\Sigma}$ is a consistent estimator of the variance–covariance matrix of $\gamma(\tau)$.

For a quantile interval $\tau \in [a, b]$, under suitable conditions and the hypothesis $H_0 : \gamma(\tau) = 0$, $\forall \tau \in \mathcal{T} \subset [a, b]$, the Wald statistic process weakly converges to

$$W_T(\tau) \Rightarrow \| \frac{\mathbf{B}_q(\tau)}{\sqrt{\tau(1-\tau)}} \|^2, \, for \, \tau \in \mathcal{T} \tag{6}$$

where $\mathbf{B}_q(\tau) = [\tau(1-\tau)]^{1/2} \mathcal{N}(0, \, I_q)$ denotes a vector of $q$ independent Brownian bridges and the weak limit is the sum of the square of $q$ independent Bessel processes. Thus, for the null hypothesis of Granger non-causality, the sup-Wald test is given by

$$\sup_{\tau \in \mathcal{T}} W_T(\tau) \to \sup_{\tau \in \mathcal{T}} \| \frac{\mathbf{B}_q(\tau)}{\sqrt{\tau(1-\tau)}} \|^2 \tag{7}$$

Following Chuang et al. [29], this study used a sequential lag selection method to determine the optimal lag truncation order. To identify the critical values of the sup-Wald test for each range and each lag order, this study generated 10,000 independent simulations using a standard Gaussian stochastic process.

## 3. Data and Descriptive Statistics

This research used weekly spot and futures prices for the maturities of one, two, three, and four months of West Texas Intermediate (WTI) (Chen et al. [28] emphasized that data frequency was very crucial and that the sampling frequency could affect the results of an empirical analysis. To test our empirical results, we used daily and monthly data frequencies in robustness checks.), also known as Texas Light Sweet, which is a type of crude oil used as the benchmark in oil pricing and is the underlying commodity of the oil futures contracts at the New York Mercantile Exchange's (NYMEX), which the market of provides important price information to buyers and sellers of crude oil around the world. This research used Spot, Contract1, Contract2, Contract3, and Contract4 to denote the WTI spot and futures prices of the contracts with varying maturities, respectively.

Figure 1 shows the WTI crude oil price trends from 2 January 1986 to 26 May 2017. After March 2003, when the Iraq War broke out, the average level and volatility of crude oil prices showed an accelerated rise. After the war, there are more occasional spikes in crude prices. Therefore, this study segmented the sample into two sampled periods: Prewar, which spans 2 January 1986 to 20 March 2003, and Postwar, which spans 21 March 2003 to 26 May 2017. Figure 2 displays the crude oil futures prices, as well as the differentials between the crude spot and futures prices. Although the overall price trends of the crude oil spot and four futures prices are similar, the differential is more significant in longer maturity pairs than in shorter maturity pairs.

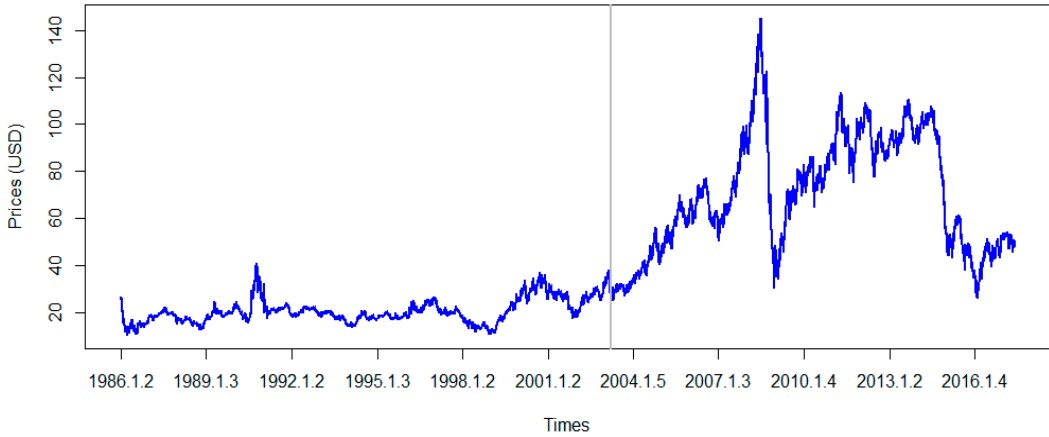

**Figure 1.** The West Texas Intermediate (WTI) crude oil spot prices.

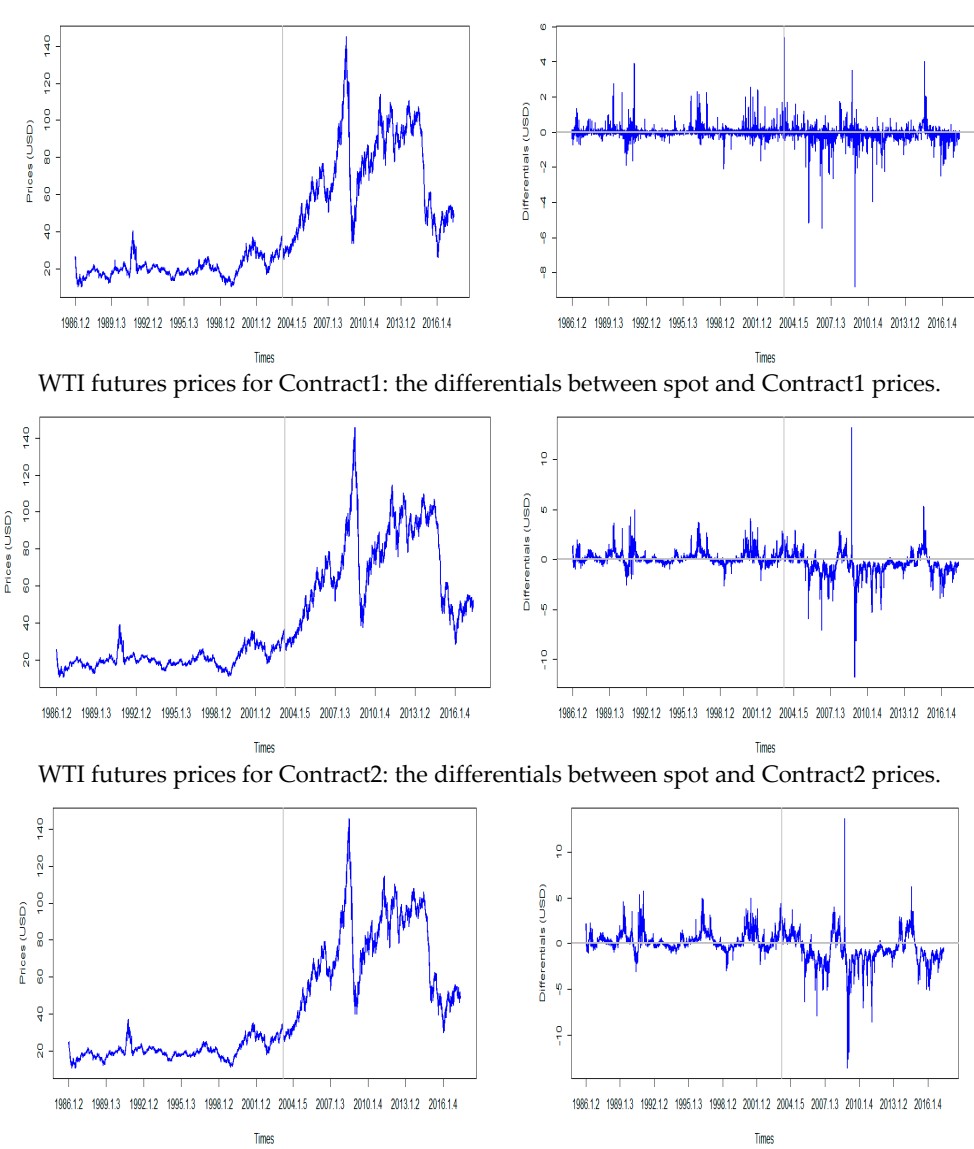

WTI futures prices for Contract1: the differentials between spot and Contract1 prices.

WTI futures prices for Contract2: the differentials between spot and Contract2 prices.

WTI futures prices for Contract3: the differentials between spot and Contract3 prices.

**Figure 2.** *Cont.*

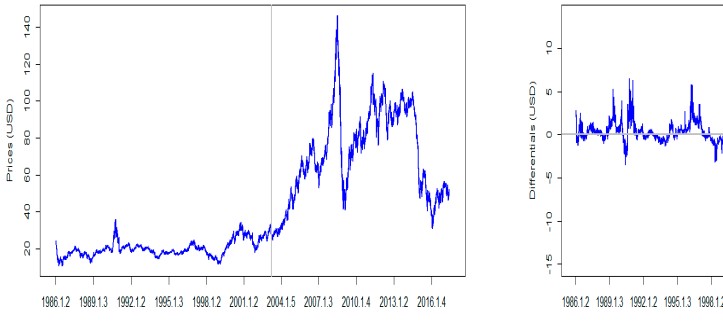

WTI futures prices for Contract4: the differentials between spot and Contract4 prices.

**Figure 2.** WTI futures prices and the differentials between spot and futures prices.

Table 1 presents key descriptive statistics for weekly crude oil spot and futures logarithmic prices. The differences between the two periods are quite evident in Table 1. For Postwar, a significant increase in the mean and variance can be observed, while a higher dispersion of the price distribution is reflected at the lower kurtosis. Additionally, the crude oil spot and futures logarithmic prices are negatively skewed for Postwar but positively skewed for Prewar. According to the Shapiro–Wilk test, the normality of the unconditional distribution was strongly rejected for all crude oil price series. To test for the presence of a unit root against the alternative of a stationary process in the price series of crude oil spot and futures prices, the Augmented Dickey–Fuller (ADF) and Kwiatkowski-Phillips-Schmidt-Shin (KPSS) tests were conducted. The ADF test offers evidence for the presence of a unit root in all crude oil price series. In the KPSS test, the null hypothesis of a stationary process is rejected for all price series. Hence, the results of the KPSS test are in line with the results of the ADF test.

**Table 1.** Descriptive statistics for the crude oil spot and futures logarithmic prices.

| | Mean | Std. Dev. | Skewness | Kurtosis | Shapiro–Wilk Test | Augmented Dickey–Fuller (ADF) Test | KPSS Test |
|---|---|---|---|---|---|---|---|
| Pre-Iraq War (2 January 1986–20 March 2003) | | | | | | | |
| Spot | 3.0021 | 0.2398 | 0.2411 | 2.9561 | 0.9848 *** | −3.1090 | 1.0077 *** |
| Contract1 | 3.0009 | 0.2390 | 0.2328 | 2.9532 | 0.9847 *** | −3.0849 | 1.0086 *** |
| Contract2 | 2.9941 | 0.2303 | 0.2359 | 2.9560 | 0.9837 *** | −3.0443 | 1.0926 *** |
| Contract3 | 2.9881 | 0.2217 | 0.2300 | 2.9341 | 0.9853 *** | −3.0606 | 1.1541 *** |
| Contract4 | 2.9825 | 0.2133 | 0.2253 | 2.9153 | 0.9835 *** | −3.0638 | 1.1971 *** |
| Post-Iraq War (21 March 2003–26 May 2017) | | | | | | | |
| Spot | 4.1785 | 0.3848 | −0.3781 | 2.1859 | 0.9604 *** | −2.6617 | 0.8450 *** |
| Contract1 | 4.1798 | 0.3841 | −0.3843 | 2.3160 | 0.9603 *** | −2.6745 | 0.8508 *** |
| Contract2 | 4.1880 | 0.3814 | −0.4416 | 2.3160 | 0.9595 *** | −2.5781 | 0.8854 *** |
| Contract3 | 4.1935 | 0.3804 | −0.4978 | 2.4259 | 0.9573 *** | −2.5200 | 0.9173 *** |
| Contract4 | 4.1972 | 0.3804 | −0.5490 | 2.5260 | 0.9545 *** | −2.4751 | 0.9463 *** |
| Full sample period (2 January 1986–26 May 2017) | | | | | | | |
| Spot | 3.5339 | 0.6644 | 0.3742 | 1.7449 | 0.9168 *** | −2.6634 | 5.3136 *** |
| Contract1 | 3.5339 | 0.6651 | 0.3733 | 1.7408 | 0.9162 *** | −2.6634 | 5.3196 *** |
| Contract2 | 3.5339 | 0.6693 | 0.3759 | 1.7084 | 0.9100 *** | −2.5169 | 5.3748 *** |
| Contract3 | 3.5331 | 0.6726 | 0.3812 | 1.6825 | 0.9041 *** | −2.4155 | 5.4146 *** |
| Contract4 | 3.5317 | 0.6752 | 0.3879 | 1.6613 | 0.8984 *** | −2.3317 | 5.4432 *** |

Notes: *** denotes a rejection at the 1% significance levels.

## 4. Empirical Results

### 4.1. The Quantile Causality Relationships between Crude Oil Spot and Futures Prices

Given the results of the unit roots, the Johansen cointegration analysis is used to test for pairwise cointegrations between spot and futures prices. The tests for the number of cointegrating vectors are based on maximal eigenvalue and trace statistics of the stochastic matrix in the multivariate framework. Table 2 presents the results of both tests, which suggest the existence of one cointegrating vector, and so, the null hypothesis of no cointegration between crude oil spot and futures prices is

rejected at the 1% significance level. Hence, we can say that each futures price maintains a long-run equilibrium relationship with a corresponding spot price. For the case when spot and futures prices are cointegrated, both may contribute to the crude oil price trend in the long term. Therefore, the analysis should be conducted within a VECM framework.

**Table 2.** The Johansen cointegration analysis for crude oil spot and futures logarithmic prices.

| Variables | Null Hypothesis | Trace Statistics | Max. Eigenvalue Statistics |
|---|---|---|---|
| Pre-Iraq War (2 January 1986–20 March 2003) | | | |
| Spot, Contract1 | None<br>At most 1 | 98.2863 (0.0001) ***<br>0.0298 (0.8878) | 98.2564 (0.0001) ***<br>0.0298 (0.8878) |
| Spot, Contract2 | None<br>At most 1 | 24.9617 (0.0003) ***<br>0.0413 (0.8677) | 24.9203 (0.0001) ***<br>0.0413 (0.8677) |
| Spot, Contract3 | None<br>At most 1 | 22.5687 (0.0007) ***<br>0.0361 (0.8764) | 22.5326 (0.0004) ***<br>0.0361 (0.8764) |
| Spot, Contract4 | None<br>At most 1 | 21.6154 (0.0011) ***<br>0.0385 (0.8722) | 21.5768 (0.0006) ***<br>0.0385 (0.8722) |
| Post-Iraq War (21 March 2003–26 May 2017) | | | |
| Spot, Contract1 | None<br>At most 1 | 71.3869 (0.0000) ***<br>0.0448 (0.8623) | 71.3421 (0.0001) ***<br>0.0448 (0.8623) |
| Spot, Contract2 | None<br>At most 1 | 32.8601 (0.0000) ***<br>0.1206 (0.7748) | 32.7395 (0.0000) ***<br>0.1206 (0.7748) |
| Spot, Contract3 | None<br>At most 1 | 26.4837 (0.0001) ***<br>0.1353 (0.7615) | 26.3483 (0.0001) ***<br>0.1353 (0.7615) |
| Spot, Contract4 | None<br>At most 1 | 22.5999 (0.0007) ***<br>0.1461 (0.7522) | 22.4537 (0.0004) ***<br>0.1461 (0.7522) |
| Full sample period (2 January 1986–26 May 2017) | | | |
| Spot, Contract1 | None<br>At most 1 | 153.2081 (0.0001) ***<br>0.0181 (0.9122) | 153.1900 (0.0001) ***<br>0.0181 (0.9122) |
| Spot, Contract2 | None<br>At most 1 | 44.3798 (0.0000) ***<br>0.0675 (0.8311) | 44.3122 (0.0000) ***<br>0.0675 (0.8311) |
| Spot, Contract3 | None<br>At most 1 | 38.6682 (0.0000) ***<br>0.1030 (0.7917) | 38.5652 (0.0000) ***<br>0.1030 (0.7917) |
| Spot, Contract4 | None<br>At most 1 | 35.4214 (0.0000) ***<br>0.1398 (0.7575) | 35.2815 (0.0000) ***<br>0.1398 (0.7575) |

Notes: *** denotes a rejection at the 1% significance levels. *p*-values are in parenthesis.

The estimation results of the VECM are shown in Table 3. The estimates indicate that only spot prices react to deviations from the cointegration relations, whereas the futures prices are exogenous with respect to the long-run relationship. This result holds for all sample periods and futures contracts even though the significance level decreases gradually as the maturity of the contract increases. Prewar, for the first pair of Spot Contract1, the spot prices react to deviations from the cointegration relation at a 1% significance level, but for the fourth pair of Spot Contract4, the significance level drops to 10%. A very consistent finding appears for the Postwar period. For the short-run relationship, there is a large distinction between the Prewar and Postwar periods. During the Prewar period, except for the first pair of Spot Contract1, crude oil futures prices significantly influenced spot prices. However, during Postwar, except for the first pair of Spot and Contract1, crude oil spot price shocks significantly influenced futures prices.

**Table 3.** Estimations of the vector error correction model (VECM).

| $i$ | | $c$ | $\beta^T p_{t-1}$ | $\Delta p_{s(t-1)}$ | $\Delta p_{f(t-1)}$ |
|---|---|---|---|---|---|
| Pre-Iraq War (2 January 1986–20 March 2003) | | | | | |
| 1 | $\Delta p_{s(t)}$ | −0.0056 (0.0021) ** | −0.9555 (0.2444) *** | 0.0645 (0.1719) | 0.0397 (0.1749) |
| | $\Delta p_{f(t)}$ | 0.0009 (0.0021) | 0.0807 (0.2415) | 0.0345 (0.1699) | 0.0465 (0.1729) |
| 2 | $\Delta p_{s(t)}$ | −0.0136 (0.0051) ** | −0.1822 (0.0634) ** | −0.1834 (0.0959) | 0.3428 (0.1089) ** |
| | $\Delta p_{f(t)}$ | −0.0001 (0.0044) | −0.0061 (0.0553) | 0.0303 (0.0838) | 0.0950 (0.0951) |
| 3 | $\Delta p_{s(t)}$ | −0.0154 (0.0061) * | −0.1079 (0.0404) ** | −0.1256 (0.0817) | 0.2959 (0.1006) ** |
| | $\Delta p_{f(t)}$ | 0.0006 (0.0049) | 0.0017 (0.0325) | 0.0097 (0.0658) | 0.1171 (0.0811) |
| 4 | $\Delta p_{s(t)}$ | −0.0171 (0.0070) * | −0.0808 (0.0316) * | −0.0717 (0.0738) | 0.2414 (0.0974) * |
| | $\Delta p_{f(t)}$ | 0.0017 (0.0053) | 0.0063 (0.0238) | 0.0156 (0.0555) | 0.1027 (0.0732) |
| Post-Iraq War (21 March 2003–26 March 2017) | | | | | |
| 1 | $\Delta p_{s(t)}$ | −0.0118 (0.0026) *** | −1.5300 (0.2616) *** | 0.1048 (0.1936) | 0.1005 (0.2012) |
| | $\Delta p_{f(t)}$ | −0.0047 (0.0026) | −0.6637 (0.2616) | 0.1404 (0.1936) | 0.0380 (0.2012) |
| 2 | $\Delta p_{s(t)}$ | −0.0137 (0.0045) ** | −0.2374 (0.0704) *** | 0.0630 (0.1262) | 0.1490 (0.1381) |
| | $\Delta p_{f(t)}$ | −0.0054 (0.0041) | −0.1027 (0.0639) | 0.3233 (0.1145) ** | −0.1392 (0.1253) |
| 3 | $\Delta p_{s(t)}$ | −0.0107 (0.0047) * | −0.1113 (0.0442) * | 0.1709 (0.1074) | 0.0221 (0.1221) |
| | $\Delta p_{f(t)}$ | −0.0020 (0.0041) | −0.0277 (0.0382) | 0.3613 (0.0929) *** | −0.1875 (0.1056) |
| 4 | $\Delta p_{s(t)}$ | −0.0086 (0.0047) | −0.0686 (0.0338) * | 0.2118 (0.0985) * | −0.0314 (0.1156) |
| | $\Delta p_{f(t)}$ | −0.0001 (0.0040) | −0.0064 (0.0282) | 0.3529 (0.0822) *** | −0.1847 (0.0965) |
| Full sample period (2 January 1986–26 March 2017) | | | | | |
| 1 | $\Delta p_{s(t)}$ | 0.0063 (0.0013) *** | −1.2848 (0.1760) *** | 0.1456 (0.1275) | 0.0024 (0.1308) |
| | $\Delta p_{f(t)}$ | 0.0020 (0.0013) | −0.3564 (0.1752) * | 0.1613 (0.1269) | −0.0362 (0.1302) |
| 2 | $\Delta p_{s(t)}$ | 0.0084 (0.0018) *** | −0.2335 (0.0437) *** | −0.0811 (0.0755) | 0.2646 (0.0842) ** |
| | $\Delta p_{f(t)}$ | 0.0038 (0.0016) * | −0.0988 (0.0389) * | 0.1584 (0.0673) * | −0.0020 (0.0751) |
| 3 | $\Delta p_{s(t)}$ | 0.0086 (0.0020) *** | −0.1316 (0.0274) *** | −0.0146 (0.0644) | 0.1959 (0.0765) * |
| | $\Delta p_{f(t)}$ | 0.0034 (0.0017) * | −0.0476 (0.0229) * | 0.1445 (0.0538) ** | 0.0082 (0.0639) |
| 4 | $\Delta p_{s(t)}$ | 0.0085 (0.0021) *** | −0.0947 (0.0212) *** | 0.0296 (0.0585) | 0.1465 (0.0731) * |
| | $\Delta p_{f(t)}$ | 0.0029 (0.0017) | −0.0285 (0.0167) | 0.1375 (0.0463) ** | 0.0094 (0.0579) |

Notes: The maturity of the futures contracts is denoted by *i*. The standard deviations are in parenthesis. *, **, and *** indicate the statistical significance at the 10%, 5%, and 1% levels, respectively.

Table 4 reports descriptive statistics of the VECM-residuals that filtered the long-run co-movements between crude oil spot and futures prices. The Ljung–Box test indicates that all residual series are not autocorrelated. The non-normal distributions for all residuals are confirmed by the Shapiro–Wilk test. The ADF and KPSS tests indicate that all VECM-residuals are stationary processes. In later studies, this study used the filtered VECM-residuals to investigate the quantile causality relationship between crude oil spot and futures prices.

**Table 4.** Descriptive statistics for VECM residuals.

|  | **Box–Ljung Test** | **Shapiro–Wilk Test** | **ADF Test** | **KPSS Test** |
|---|---|---|---|---|
| Pre-Iraq War (2 January 1986–20 March 2003) | | | | |
| Spot | 0.0004 | 0.9551 *** | −9.5090 *** | 0.1010 |
| Contract1 | 0.0018 | 0.9528 *** | −9.3631 *** | 0.1077 |
| Contract2 | 0.0001 | 0.9543 *** | −9.3493 *** | 0.1013 |
| Contract3 | 0.0038 | 0.9508 *** | −9.2140 *** | 0.1015 |
| Contract4 | 0.0069 | 0.9455 *** | −9.0654 *** | 0.1050 |
| Post-Iraq War (21 March 2003–26 March 2017) | | | | |
| Spot | 0.0133 | 0.9767 *** | −7.1097 *** | 0.3090 |
| Contract1 | 0.0429 | 0.9796 *** | −7.1769 *** | 0.2459 |
| Contract2 | 0.2301 | 0.9887 *** | −7.1594 *** | 0.3296 |
| Contract3 | 0.2021 | 0.9893 *** | −7.2937 *** | 0.3358 |
| Contract4 | 0.1770 | 0.9883 *** | −7.3710 *** | 0.3340 |
| Full sample period (2 January 1986–26 March 2017) | | | | |
| Spot | 0.0107 | 0.9619 *** | −11.7842 *** | 0.0978 |
| Contract1 | 0.0314 | 0.9624 *** | −12.1272 *** | 0.0685 |
| Contract2 | 0.0387 | 0.9704 *** | −11.8287 *** | 0.0775 |
| Contract3 | 0.0127 | 0.9709 *** | −11.6817 *** | 0.0835 |
| Contract4 | 0.0039 | 0.9693 *** | −11.5503 *** | 0.0895 |

Notes: The Box–Liung test for autocorrelation of VECM residuals, the Shapiro–Wilk test for the normality of VECM residuals, and the ADF test and KPSS test for unit roots of VECM residuals. *** denotes a rejection at the 1% significance levels.

Table 5 shows the empirical results for the Granger non-causality in the mean tests for crude oil spot and futures prices. The classical non-causality test proposed by Granger [35] identifies the causal relationship on the basis of the estimated conditional mean behavior. This empirical evidence indicates no significant conditional mean causal relationships between crude oil spot and futures prices because the VECM model has already purged the residuals of linear dependence, but the classical Granger non-causality test only uncovered the linear causal relationship on average. To uncover a more complete causal relationship between crude oil spot and futures prices, this study employed a quantile causality test to explore the causal relationships over different levels of conditional quantiles of the dependent variable.

**Table 5.** The test for the Granger causality in the mean for crude oil spot and futures prices.

|  | **Spot → Futures** | | **Futures → Spot** | |
|---|---|---|---|---|
|  | ***p*-Value** | **Causality** | ***p*-Value** | **Causality** |
| Pre-Iraq War (2 January 1986–20 March 2003) | | | | |
| Contract1 | 0.9248 (3) | No | 0.9668 (3) | No |
| Contract2 | 0.7539 (5) | No | 0.5607 (5) | No |
| Contract3 | 0.9589 (2) | No | 0.7166 (2) | No |
| Contract4 | 0.9782 (3) | No | 0.8439 (3) | No |
| Post-Iraq War (21 March 2003–26 March 2017) | | | | |
| Contract1 | 0.8932 (7) | No | 0.8485 (7) | No |
| Contract2 | 0.4026 (7) | No | 0.1698 (7) | No |
| Contract3 | 0.6034 (3) | No | 0.3245 (3) | No |
| Contract4 | 0.6993 (2) | No | 0.4058 (2) | No |

Notes: The table reports the results for the null hypothesis of Granger non-causality in the means. *p*-values for the null hypothesis are reported, with the numbers in brackets indicating the lag order based on the Akaike information criterion (AIC).

Table 6 reports the results of the sup-Wald statistic and the corresponding truncation order for causality between spot and futures crude oil prices before the Iraq War. Regarding causality from futures to spot prices, these results revealed causal relationships at the 1% significance level for all

quantile intervals except for the middle quantile levels [0.2 0.5] and [0.5 0.8]. For the middle intervals, there is causality from futures to spot crude oil prices only in the quantile interval [0.2 0.5] for Contract 4 and in [0.5 0.8] for Contract 2 at the 10% significance level. These results show that there is a strong causal relationship over the tail region of the conditional distribution and that the causal relationship over the middle interval is weaker. This implies that when the performances of crude oil markets are in extreme situations, shocks of futures prices can significantly cause an impact on spot prices, but when the crude oil prices fluctuate near their medians, the futures prices have no significant effects on the spot prices. As for reverse causality, this study finds that the crude oil spot price Granger-caused futures prices at the 10% significance level only in quantile intervals [0.05 0.2] and [0.05 0.5] for Contract1 and Contract2, implying that crude oil spot prices had causal effects on futures prices at the lower quantiles for short-term contracts but not in the intermediate and upper quantiles for longer-dated futures. Not surprisingly, futures oil prices of short maturities contained more complete available information than did the futures prices of long maturities. Especially when the performance of the spot market was in the worst situation, short-term investors, such as noise traders and speculators, always traded with a large volume and displayed greater and faster responsiveness to new information in the oil spot market.

**Table 6.** The quantile causality results for crude oil spot and futures prices: Pre-Iraq War.

| | Spot → Futures | | | | Futures → Spot | | | |
|---|---|---|---|---|---|---|---|---|
| | Contract1 | Contract2 | Contract3 | Contract4 | Contract1 | Contract2 | Contract3 | Contract4 |
| [0.05 0.2] | 9.2374 ** [2] | 8.3121 * [2] | 2.0839 [1] | 2.2403 [2] | 9.4682 ** [2] | 18.2862 *** [2] | 17.1607 *** [2] | 19.9029 *** [2] |
| [0.05 0.5] | 9.2374 ** [2] | 8.3121 * [2] | 2.0839 [1] | 2.0627 [2] | 9.1795 ** [2] | 17.4017 *** [2] | 16.9700 *** [2] | 19.6770 *** [2] |
| [0.05 0.95] | 6.2136 [2] | 6.3121 [2] | 1.8371 [2] | 1.9383 [2] | 23.4537 *** [2] | 29.7522 *** [2] | 16.9700 *** [2] | 19.2939 *** [2] |
| [0.2 0.5] | 4.4442 [2] | 2.6224 [2] | 1.8447 [2] | 2.0627 [2] | 4.0405 [2] | 4.7525 [1] | 5.6666 [2] | 8.7082 * [2] |
| [0.2 0.95] | 4.6850 [2] | 3.4934 [1] | 1.8242 [2] | 1.9994 [2] | 23.4537 *** [2] | 29.4822 *** [2] | 14.8924 *** [2] | 18.7581 *** [2] |
| [0.5 0.8] | 4.8203 [2] | 2.4140 [1] | 1.3420 [1] | 1.0846 [2] | 7.3793 * [2] | 9.4254 ** [2] | 4.0643 [2] | 3.8276 [2] |
| [0.5 0.95] | 4.8203 [2] | 3.4690 [1] | 1.3378 [1] | 1.3442 [2] | 23.4537 *** [2] | 29.7522 *** [2] | 14.6251 *** [2] | 19.5730 *** [2] |
| [0.8 0.95] | 4.2922 [2] | 3.4934 [1] | 0.9152 [1] | 1.5146 [2] | 23.4537 *** [2] | 30.4697 *** [2] | 14.9743 *** [2] | 28.3354 *** [2] |

Notes: The sup-Wald test statistics and the selected lag order (in square brackets) are reported. *, **, and *** indicate the statistical significance at the 10%, 5%, and 1% levels, respectively.

Table 7 reveals the results for quantile causality between crude oil spot and futures prices after the Iraq War. The sup-Wald statistics support the evidence that the causal relationships between crude oil spot and futures prices are heterogeneous across quantiles of price distributions. Regarding the causality from crude oil futures prices to spot price, there is no causal relationship over the middle quantile levels [0.2 0.5] and [0.5 0.8] for all mature futures contracts, implying that there is causality from futures to spot prices for only a low or high level of crude oil prices. At the lower quantile levels [0.05 0.2] and [0.05 0.5], crude oil futures prices Granger-caused spot prices at a 10% significance level only for Contract1. However, there is causality from futures to spot crude oil prices at a 1% significance level at the upper quantile levels [0.5 0.95] and [0.8 0.95] for Contract2, Contract3, and

Contract4. The heterogeneous causal relationships between crude oil spot and futures prices should be ascribed to the adjustments on the impacts of futures prices of long maturities on spot prices according to the performances of the crude oil markets. When the crude oil markets are in the worst situation, the short-term investors expect futures prices to show a sign of reverting. However, when the crude oil markets perform better, the optimistic sentiment will drive long-term investors to expect better performances in the spot markets. As for reverse causality, this study finds that crude oil spot prices Granger-caused futures prices at the upper quantile levels [0.5 0.95] and [0.8 0.95] at the 1% significance level, implying that the causal impact of crude oil spot prices on futures prices occurred at the upper quantiles but not at the lower and intermediate quantiles, i.e., when spot markets perform better, the information on spot prices is useful in anticipating futures prices.

**Table 7.** The quantile causality results for crude oil spot and futures prices: Post-Iraq War.

| | Spot → Futures | | | | Futures → Spot | | | |
| --- | --- | --- | --- | --- | --- | --- | --- | --- |
| | Contract1 | Contract2 | Contract3 | Contract4 | Contract1 | Contract2 | Contract3 | Contract4 |
| [0.05 0.2] | 4.5930 [2] | 5.3037 [1] | 6.5114 [1] | 7.6346 * [1] | 8.4553 * [2] | 3.9274 [1] | 4.0869 [2] | 3.7192 [1] |
| [0.05 0.5] | 4.3737 [2] | 5.3037 [1] | 6.5114 [1] | 7.6346 * [1] | 8.4499 * [2] | 3.9279 [1] | 4.0869 [2] | 6.2898 [2] |
| [0.05 0.95] | 4.3737 [2] | 34.1576 *** [2] | 23.2410 *** [2] | 13.0549 ** [2] | 8.4499 * [2] | 4.05337 [2] | 20.4756 *** [1] | 11.6323 ** [1] |
| [0.2 0.5] | 3.0623 [2] | 5.0740 [2] | 1.9048 [1] | 0.9275 [1] | 3.4061 [2] | 0.7471 [2] | 2.6708 [2] | 3.6211 [2] |
| [0.2 0.95] | 3.0885 [2] | 32.8537 *** [2] | 22.6714 *** [2] | 6.3535 [2] | 2.6947 [1] | 39.5296 *** [2] | 21.4325 *** [1] | 11.6323 ** [1] |
| [0.5 0.8] | 1.9475 [1] | 4.6018 [1] | 3.4892 [1] | 1.8463 [1] | 2.7098 [1] | 1.1411 [1] | 1.3939 [2] | 1.3927 [2] |
| [0.5 0.95] | 3.0885 [2] | 34.1576 *** [2] | 23.2411 *** [2] | 13.0549 *** [2] | 2.7058 [1] | 40.5337 *** [2] | 22.8051 *** [1] | 17.1111 *** [1] |
| [0.8 0.95] | 3.0885 [2] | 34.2209 *** [2] | 23.6555 *** [2] | 13.0649 *** [2] | 2.2274 [2] | 41.0123 *** [2] | 23.1001 *** [1] | 17.3480 *** [2] |

Notes: The sup-Wald test statistics and the selected lag order (in square brackets) are reported. *, **, and *** indicate the statistical significance at the 10%, 5%, and 1% levels, respectively.

By comparing the results of the causality in the mean and quantile causality, this study reveals three important findings: (a) the causal relationships between crude oil spot and futures prices show a heterogeneous behavior. Before the Iraq War, crude oil spot and futures prices were mutually Granger-caused at the lower quantile levels, indicating that both spot and futures prices provided useful information for forecasting the prices. However, at the upper quantile levels, the futures prices drove spot prices to the equilibrium level but not vice versa. After the Iraq War, in the higher quantile regions, the results reveal a clear bidirectional causality, i.e., both spot and futures prices react simultaneously to new information. However, when the oil markets are in the worst situation, only the causality from futures to spot crude oil prices exists but not vice versa. The heterogeneous behaviors can provide more useful information to investors for hedging and investing in different market situations. (b) The causal relationships between the crude oil spot and futures prices present an obvious tail effect, i.e., either for the causal relationships from crude futures prices to spot prices or for reverse causality. The causality exists only over the lower and upper quantile intervals but not the middle quantile levels, implying a causality between crude oil spot and futures prices only for a high or low level of oil prices. Therefore, we can say that, when the oil prices fluctuate near their medians, the causal relationships between spot and futures crude oil prices are weaker. (c) The causal

relationships between crude oil spot and futures prices before and after the Iraq War are distinctly different. Regarding the causal relationships from crude oil futures prices to spot prices, before the Iraq War, there was significant causality over the lower and upper quantile intervals for all mature futures contracts. However, after the war, the causality existed only at the lower quantile levels for short-term contracts and at the upper quantile levels for longer-term contracts. Regarding the causal relationships from spot to futures crude oil prices, there is causality only at the lower quantile levels for short-term contracts before the war and only at upper quantile levels for longer-term contracts after the war. The different causalities before and after the Iraq War may be explained by the fact that the international crude oil markets faced more volatility after the war and stimulated different speculative and arbitrage behaviors.

## 4.2. Robustness Checks

Data frequency is very crucial according to Westgaard et al. [11], who emphasized that the sampling frequency could affect the results of the empirical analysis, and according to Wang and Wu [12], who found that different data frequencies would induce an economic necessity to investigate the relationships between spot and futures price series. Therefore, this study checked if the results were confirmed by using different data frequencies, including daily and monthly data.

Tables 8 and 9 show the results of the robustness check with daily data. These sup-Wald statistics support the evidence that the causal relationships between crude oil spot and futures prices present distinct tail effects and heterogeneity. These findings are almost consistent with the results obtained with weekly data, which indicate that these main findings reported above do not seem to be sensitive to different data frequencies.

**Table 8.** A robustness check: the quantile causality results for crude oil spot and futures prices using daily data: Pre-Iraq War.

| | Spot → Futures | | | | Futures → Spot | | | |
|---|---|---|---|---|---|---|---|---|
| | **Contract1** | **Contract2** | **Contract3** | **Contract4** | **Contract1** | **Contract2** | **Contract3** | **Contract4** |
| [0.05 0.2] | 15.4848 *** [2] | 10.8379 ** [2] | 5.4126 [2] | 6.3391 [2] | 25.8221 *** [1] | 22.7705 *** [1] | 16.1468 *** [1] | 34.8730 *** [2] |
| [0.05 0.5] | 14.6070 *** [1] | 7.4860 * [2] | 4.9811 [2] | 5.4934 [2] | 24.7296 *** [1] | 27.1477 *** [1] | 14.5048 *** [1] | 29.1598 *** [2] |
| [0.05 0.95] | 4.6070 [1] | 10.6821 ** [2] | 18.7855 *** [1] | 23.6111 *** [1] | 24.5047 *** [1] | 25.1895 *** [1] | 14.2752 *** [1] | 29.1598 *** [2] |
| [0.2 0.5] | 4.6885 [1] | 2.7414 [1] | 2.3496 [1] | 2.6201 [1] | 4.5315 [2] | 3.4653 [2] | 4.5449 [2] | 2.5674 [1] |
| [0.2 0.95] | 4.6070 [1] | 10.6821 ** [2] | 17.3635 *** [1] | 26.2982 *** [1] | 14.2069 *** [2] | 15.5510 *** [1] | 16.1358 *** [2] | 14.2302 *** [1] |
| [0.5 0.8] | 0.9869 [1] | 4.3991 [2] | 2.3702 [2] | 2.4702 [1] | 8.2765 * [2] | 5.4818 [1] | 6.0435 [2] | 4.2267 [1] |
| [0.5 0.95] | 2.5891 [2] | 10.6821 ** [2] | 4.1296 [2] | 7.8052 * [1] | 14.0264 *** [2] | 15.2937 *** [1] | 15.8470 *** [1] | 14.2169 *** [2] |
| [0.8 0.95] | 1.4630 [2] | 10.6821 ** [2] | 3.4805 [2] | 8.1365 * [1] | 14.9434 *** [2] | 14.6544 *** [1] | 11.6118 ** [1] | 12.0608 ** [1] |

Notes: The sup-Wald test statistics and the selected lag order (in square brackets) are reported. *, **, and *** indicate the statistical significance at the 10%, 5%, and 1% levels, respectively.

**Table 9.** A robustness check: the quantile causality results for crude oil spot and futures prices using daily data: Post-Iraq War.

| | Spot → Futures | | | | Futures → Spot | | | |
|---|---|---|---|---|---|---|---|---|
| | **Contract1** | **Contract2** | **Contract3** | **Contract4** | **Contract1** | **Contract2** | **Contract3** | **Contract4** |
| [0.05 0.2] | 5.9879 [2] | 4.2671 [2] | 4.2104 [2] | 9.8374 * [2] | 60.2010 *** [2] | 29.6089 *** [2] | 26.0349 *** [2] | 5.7807 [1] |
| [0.05 0.5] | 5.7533 [2] | 4.2671 [2] | 3.2104 [2] | 8.1743 * [2] | 60.2010 *** [2] | 29.6089 *** [2] | 25.9487 *** [2] | 5.6538 [1] |
| [0.05 0.95] | 5.4563 [2] | 8.5152 * [2] | 9.2749 * [2] | 5.5018 [2] | 57.8418 *** [2] | 29.0009 *** [2] | 24.7870 *** [2] | 24.7957 *** [2] |
| [0.2 0.5] | 2.4960 [2] | 5.3953 [2] | 6.7061 [2] | 9.4760 ** [2] | 7.0655 [2] | 8.6087 [2] | 5.0962 [2] | 4.0469 [2] |
| [0.2 0.95] | 7.1707 [1] | 4.3990 [2] | 15.0111 *** [2] | 9.3666 ** [2] | 6.1943 [2] | 17.8461 [2] | 18.2479 [2] | 6.7962 [2] |
| [0.5 0.8] | 11.0982 ** [2] | 12.7659 ** [1] | 4.1307 [2] | 14.4846 *** [2] | 8.4318 * [2] | 11.0112 ** [2] | 6.2218 [2] | 5.0121 [2] |
| [0.5 0.95] | 8.3867 [1] | 4.4475 [1] | 15.0111 *** [2] | 14.4846 *** [2] | 5.4098 [2] | 19.8665 *** [2] | 18.2479 *** [2] | 6.7962 [2] |
| [0.8 0.95] | 8.7437 [1] | 4.4668 [1] | 15.0111 *** [2] | 13.0619 *** [2] | 3.0464 [1] | 19.9091 *** [2] | 19.3863 *** [2] | 7.3149 [2] |

Notes: The sup-Wald test statistics and the selected lag order (in square brackets) are reported. *, **, and *** indicate the statistical significance at the 10%, 5%, and 1% levels, respectively.

In the same way, Tables 10 and 11 reveal the sup-Wald statistic and the corresponding truncation order for monthly data. In fact, because of the existing heterogeneous behaviors between short-term and long-term investors (Short-term investors, such as noise traders and speculators, concentrate on intraday, daily, or weekly oil price changes and may even trade with a large volume when a minor oil price change occurs. However, long-term investors, such as government and oil producers, focus on quarterly or yearly price dynamics and may trade only when oil prices adequately change over a long period.), we can see that crude oil spot and futures price relationships are not entirely consistent at different data frequencies. By and large, however, the results from the robustness checks with monthly data largely support the evidence of significant tail effects and heterogeneous behaviors for a causal relationship between crude oil spot and futures prices. In addition, it is worth pointing out that the significance of the causal relationship is very weak for monthly data. A plausible explanation is that the arbitrage activity related to monthly data in the medium term is very low and that the change of fundamentals had not been completed within a month, thereby resulting in weaker causal behaviors.

**Table 10.** A robustness cheek: the quantile causality results for crude oil spot and futures prices using monthly data: Pre-Iraq War.

| | Spot → Futures | | | | Futures → Spot | | | |
|---|---|---|---|---|---|---|---|---|
| | Contract1 | Contract2 | Contract3 | Contract4 | Contract1 | Contract2 | Contract3 | Contract4 |
| [0.05 0.2] | 8.3865 * [1] | 8.0887 * [1] | 5.1641 [1] | 4.9399 [1] | 9.3865 ** [1] | 8.0932 * [1] | 7.9731 * [1] | 4.6704 [1] |
| [0.05 0.5] | 9.3616 ** [1] | 5.0887 [1] | 7.7232 * [1] | 8.8796 * [1] | 16.3616 *** [1] | 9.9604 ** [1] | 7.6927 * [1] | 4.1412 [1] |
| [0.05 0.95] | 8.1970 * [1] | 9.0991 * [1] | 8.0268 * [1] | 4.7779 [1] | 7.1970 * [1] | 8.3519 * [1] | 10.6387 ** [1] | 7.4150 * [1] |
| [0.2 0.5] | 1.6374 [2] | 4.1192 [2] | 1.8168 [2] | 0.9915 [1] | 1.5952 [2] | 5.6270 [2] | 3.5611 [2] | 0.7510 [1] |
| [0.2 0.95] | 7.7873 * [2] | 7.7286 * [2] | 4.6896 [2] | 1.8588 [1] | 7.8873 * [1] | 8.3848 * [2] | 4.3878 [2] | 4.7086 [2] |
| [0.5 0.8] | 4.1830 [1] | 1.2782 [1] | 2.5044 [2] | 1.3040 [1] | 9.1830 ** [1] | 8.0836 * [1] | 0.8384 [1] | 1.6212 [2] |
| [0.5 0.95] | 5.1970 [1] | 2.4625 [1] | 4.6200 [2] | 1.8588 [1] | 9.1970 ** [1] | 8.3372 * [1] | 4.42106 [2] | 7.4150 * [2] |
| [0.8 0.95] | 5.2425 [1] | 4.7286 [1] | 4.6896 [2] | 1.9002 [1] | 17.2425 *** [1] | 12.3644 *** [1] | 9.8822 ** [2] | 7.4532 * [2] |

Notes: The sup-Wald test statistics and the selected lag order (in square brackets) are reported. *, **, and *** indicate the statistical significance at the 10%, 5%, and 1% levels, respectively.

**Table 11.** A robustness cheek: the quantile causality results for crude oil spot and futures prices using monthly data: Post-Iraq War.

| | Spot → Futures | | | | Futures → Spot | | | |
|---|---|---|---|---|---|---|---|---|
| | Contract1 | Contract2 | Contract3 | Contract4 | Contract1 | Contract2 | Contract3 | Contract4 |
| [0.05 0.2] | 4.7512 [2] | 5.1152 [1] | 5.2672 [1] | 6.4581 [1] | 9.1397 ** [2] | 7.4197 * [2] | 8.3580 * [1] | 8.4836 * [2] |
| [0.05 0.5] | 5.7512 [2] | 5.0831 [1] | 3.2672 [1] | 5.3802 [1] | 8.9742 * [2] | 7.4197 * [2] | 8.3580 * [1] | 8.4836 * [2] |
| [0.05 0.95] | 5.5408 [2] | 4.9886 [1] | 4.5152 [1] | 6.2128 [1] | 8.9742 * [2] | 6.7313 [2] | 7.8840 * [1] | 8.2559 * [2] |
| [0.2 0.5] | 1.2418 [1] | 1.8842 [1] | 3.9127 [2] | 3.5803 [2] | 1.2244 [1] | 2.2678 [2] | 2.4113 [2] | 1.9993 [2] |
| [0.2 0.95] | 6.5433 [2] | 1.8842 [1] | 3.9127 [1] | 4.4660 [2] | 7.2887 * [2] | 2.2624 [2] | 2.4113 [1] | 2.9429 [2] |
| [0.5 0.8] | 5.5519 [1] | 8.5700 * [1] | 7.7447 * [1] | 1.7811 [2] | 1.4503 [1] | 1.6580 [1] | 1.4408 [1] | 1.5228 [1] |
| [0.5 0.95] | 6.7397 [2] | 8.7412 * [1] | 9.5149 ** [1] | 8.4168 * [2] | 7.2887 * [2] | 8.7569 * [1] | 8.8760 * [2] | 1.5228 [1] |
| [0.8 0.95] | 6.7397 [2] | 8.7547 * [1] | 11.5755 ** [1] | 10.4434 ** [2] | 7.2887 * [2] | 9.7569 * [1] | 1.4160 [2] | 7.9429 * [2] |

Notes: The sup-Wald test statistics and the selected lag order (in square brackets) are reported. *, **, and *** indicate the statistical significance at the 10%, 5%, and 1% levels, respectively.

## 5. Conclusions

In crude oil spot futures markets, different market conditions lead to various investor behaviors and eventually bring about heterogeneous causal relationships in booming and depressing markets. This study investigated the causal relationships between crude oil spot and futures prices from the perspective of quantile causality in which the lower quantile levels represent the depressing market conditions and the upper quantile levels represent the booming market conditions. The quantile

causality test allows us to capture the causal relationships over different levels of conditional quantiles of the dependent variable, especially for extreme upper and lower quantile regions.

Using a two-stage quantile causality test method applied to crude oil spot and futures prices for the period 2 January 1986 to 26 May 2017, this study finds heterogeneous evidence regarding causal relationships between crude oil spot and futures prices, i.e., the causal relationships are different in booming or depressing market conditions. The results of the quantile causality test indicate that the significant causal relationships between crude oil spot and futures prices derive from tail quantile intervals. Moreover, by dividing the sample period into two separate periods, namely Prewar and Postwar, this study found causalities from futures to spot prices at the lower and upper quantile levels both before and after the Iraq War. However, the causality from spot to futures prices is significant only at lower quantile levels before the Iraq War and only at upper quantile levels after the Iraq War.

The results of this study indicate that the causal relationships between crude oil spot and futures prices display significant heterogeneous effects and occur only in specific quantile levels. Thus, we hope that these empirical results could provide more useful information to investors who want to hedge or manage downside or upside risks in their portfolios by using crude oil spot or futures prices. If the investors want to hedge risk by futures contracts, then they should consider the current market conditions to avoid imprecise predictions. In a further study, we should focus on the transmission mechanisms underlying the causalities discovered in the quantiles and should identify the determinants of such causalities.

**Author Contributions:** X.S. designed the main stages of the research, acquired and analyzed the data, and drafted the initial manuscript. H.Z. designed the study and was involved in drafting the initial manuscript. X.Y. revised the manuscript critically for important content. All authors have read and approved the final manuscript.

**Funding:** This research was supported by the National Natural Science Foundation of China under Grants No. 71671062, the Youth Science and Technology Talent Growth Project of the Guizhou Provincial Department of Education under Grant No. [2018]154, and the Introducing Talent Research Project of the Guizhou University of Finance and Economics in 2017

**Acknowledgments:** 

**Conflicts of Interest:** The authors declare no conflict of interest.

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
