# Peer review of "Heterogeneous Causal Relationships between Spot and Futures Oil Prices: Evidence from Quantile Causality Analysis"

_sustainability, doi:10.3390/su11051359_

Round 1
Reviewer 1 Report
Some rewording for better understanding is recommended.
Author Response
Detailed Response to the Reviewer 1
Dear Reviewer,
Thank you very much for reviewing the revised manuscript. We believe that your comments and suggestions are highly constructive and very helpful for our research. Thanks very much again for your detailed and thorough review. We have thoughtfully taken into your comments and responded to your constructive suggestions from point by point outlined below. We hope we have addressed all of your concerns.
For the sake of presentation, the comments of the referee are numbered and duplicated in italics, and our responses are given in plain. The page and line numbers of revised texts in our responses refer to our revised manuscript.
1. Some rewording for better understanding is recommended.
Responses
Thank you very much for your constructive comments. We have carefully checked the spelling and grammatical errors and used the English language editing service of MogoEdit to polish our paper. In addition, we have rewritten the Introduction section and added more discussion on the motivation and contributions of our paper.

Reviewer 2 Report
The methodology used in this paper has a critical problem. It is quite well known that the spot and futures prices are cointegrated. So the authors should consider such information in the conditional quantile non-causality test. Using the residuals from the VECM estimates quite misleads the purpose of this paper. All useful information should be reflected when one wants to test non-causality. I recommend the authors to use quantile cointegrating regression by Xiao (2009) to deal with such a dynamic structure. Moreover, in the empirical results, the authors had better illustrate the findings of the paper instead of stating the test statistics numbers.
Author Response
Detailed Response to Reviewer 2
Dear Reviewer,
Thank you very much for reviewing the revised manuscript. We believe that your comments and suggestions are highly constructive and very helpful for our research. Thanks very much again for your detailed and thorough review. We have thoughtfully taken into your comments and responded to your constructive suggestions from point by point outlined below. We hope we have addressed all of your concerns.
For the sake of presentation, the comments of the referee are numbered and duplicated in italics, and our responses are given in plain. The page and line numbers of revised texts in our responses refer to our revised manuscript.
1. The methodology used in this paper has a critical problem. It is quite well known that the spot and futures prices are cointegrated. So the authors should consider such information in the conditional quantile non-causality test. Using the residuals from the VECM estimates quite misleads the purpose of this paper. All useful information should be reflected when one wants to test non-causality. I recommend the authors to use quantile cointegrating regression by Xiao (2009) to deal with such a dynamic structure.
Responses
Thank you very much for these constructive and insightful comments. As you said, the crude oil spot and futures prices are cointegrated, so we should consider such information in the conditional quantile non-causality test. In our study, we used a vector error correction model (VECM) to filter the long-run comovements in the level of prices and we think that’s feasible. The VECM has been widely used in empirical studies to filter price series from long-run comovements (Hammoudeh et al., 2004; Bekiros and Diks, 2008; Kawamoto and Hamori, 2011; Serra et al., 2010; Hassouneh et al., 2012; Bekiros, 2014; Cabrera and Schulz, 2016). In particular, Bekiros and Diks (2008) examined the nonlinear causal relationships of VECM filtered residuals in study the nonlinear causal relationships between crude oil spot and futures prices. They pointed out that testing the filtered VECM-residuals can ensure that any causality found is strictly nonlinear in nature.
In pages 3 of our revised manuscript, we have reworded the corresponding sentence as follows:
To capture the heterogeneous causal relationships across different market conditions, we applied a two-stage approach to characterize the causal relationships between crude oil spot and futures prices. In the first step, we used a vector error correction model (VECM) to filter the long-run comovements in the level of prices. This is a reasonable setting for several reasons. On the one hand, it is quite well known that the spot and futures prices are cointegrated, so it is necessary to control the long-run comovement effects because they affect the specifications of the model used for causality testing. If the long-run comovement is not modeled, the evidence may vary significantly toward detecting linear and nonlinear causalities between the predictor variables (Bekiros and Diks, 2008). On the other hand, the VECM has been widely used in empirical studies to filter price series from long-run comovements (Hammoudeh et al., 2004; Bekiros and Diks, 2008; Kawamoto and Hamori, 2011; Serra et al., 2010; Hassouneh et al., 2012; Bekiros, 2014; Cabrera and Schulz, 2016).
Moreover, in pages 4 of our revised manuscript, we have discussed the differences between our proposed two-step method and the previous models as follows:
Our proposed two-stage causal analysis framework is one of the contributions of this paper to the existing literature. The previous models that took the market’s effects on the crude oil spot-futures relationships into account include mainly the structural break (Maslyuk and Smyth, 2009; Arouri et al., 2012; Chen et al., 2014), multivariate threshold regression (Huang et al., 2009; Wang and Wu, 2013), and quantile cointegrating models (Lee and Zeng, 2011). However, although the structural breaks and threshold methods can be used to explore the crude oil spot-futures relationships among special market states, how the causal relationships between spot and futures prices change across the quantiles of price distribution is little known. The quantile cointegration method allows for quantile-varying in a cointegrating vector and can offer much information for an understanding of the nonlinear relationships in various market conditions. However, the quantile cointegrating regressions can estimate the long-run relations between spot and future oil prices conditional on market innovations but cannot explain the short-term dynamic relationships under various market conditions. Therefore, we employed a VECM together with a quantile causality test to analyze the causal relationships between spot and futures prices on the basis of the performances of the crude oil markets. The two-stage model provides a flexible procedure and expands the available information set by considering long-run comovements and capturing various market states.
According to your suggestion, we plan to use the quantile cointegrationg regression to examine the relationship between crude oil spot and futures prices. Different from the quantile cointegration estimation technique in a static regression proposed by Xiao (2019), we use the dynamic QARDL-ECM (Quantile autoregressive distributed lag error correction model) proposed by Cho et al., (2015) to address the long-run and short-run relationships between crude oil spot and futures prices. The QARKL-ECM (p, q) model specification as
(1)
Where denotes the change in the crude oil spot price series from time to time , denotes the change in the crude oil spot price series from time to time , the parameter measuring the degree of dividend smoothing, the parameter measuring the long-run cointegrating, the parameter measuring the cumulative impact of past spot prices growth on the current spot prices growth, the parameter measuring the cumulative impact of past futures prices growth on the current spot prices growth. For examining the effects of crude spot prices on futures prices, we also use the QARKL-ECM (p, q) model specification as
. (2)
We employ the QARKL-ECM (p, q) model to study the relationships between crude oil spot and futures prices for the full sample period, Pre the Iraq War and Post the Iraq War periods.
However, we apologize for not being able to do this programmatically due to shortage of time. In the following work, we will continue to complete this work. My email is [email protected]. I am looking forward to getting your contact information and I will email you as soon as I have the results.
We hope we have addressed your concerns and the revision could fulfill your requirements. Thanks again for your comments.
2. Moreover, in the empirical results, the authors had better illustrate the findings of the paper instead of stating the test statistics numbers.
Responses
Thank you very much for your valuable comments. Our study has three important findings: (1) we find that the causal relationships between crude oil spot and futures prices show a heterogeneous behavior under different market conditions, (2) we find that the causal relationships between the crude oil spot and futures prices present an obvious tail effects, (3) we find that the causal relationships between crude oil spot and futures prices before and after the Iraq War are distinctly different. To better illustrate the findings, we have highlighted these findings in the revision.
We hope we have addressed your concerns and the revision could fulfill your requirements. Thanks again for your comments.
In pages 4-5 of our revised manuscript, we have reworded the statements as follows:
In our empirical analysis, apart from the heterogeneous causal relationships between crude oil spot and futures prices across extreme booming and depressing market conditions, we also tested the causality relationships between crude oil spot and futures prices before and after the onset of the Iraq War. The key findings of our paper are as follows.
First, we provide new evidence that the causal relationships between crude oil spot and futures prices present significant heterogeneous behaviors under different market conditions. We have observed that the causal relationships between crude oil spot and futures prices over only the lower and upper quantile intervals but not the middle quantile intervals. The causal relationships between crude oil spot and futures prices are weaker when oil prices fluctuate at nearly their medians, and are highly likely to be present when one market shows very good or very poor performance. This empirical result expands on and refines the findings of Huang et al. (2009), as well as those of Wang and Wu (2013), who suggested that at least one causal relationship between spot and futures prices exists only when the price differentials are larger than a threshold value. Our results were able to capture the causal relationships between crude oil spot and futures prices at specified quantile levels. Moreover, as compared to the conclusion of Lee and Zeng (2011), we found that crude oil spot and futures prices were influenced by quantiles and futures contracts, but they had found that spot oil prices indeed caused futures oil prices. This finding was not apparent in our empirical results. The difference in the sample interval we had selected may be the underlying reason for the inconsistency.
Second, we have revealed the nonlinear causality between crude oil spot and futures prices from the perspectives of different market states. We found that the direction and significance of causal relationships between crude oil spot and futures prices changed with different market conditions. This result is in accord with Bekiros and Diks (2008), who suggested that if nonlinear effects were accounted for, neither market led or lagged behind the other consistently. Chang and Lee (2015) used a wavelet coherency method and studied the time-varying causal relationships between crude oil spot and futures prices. Their results show significant dynamic causality between variables in the time-frequency domain. Here, we consider the nonlinear drivers, which are different from the focus of previous studies, and obtain a consistent conclusion.
Third, we observed that the causal relationships between crude oil spot and futures prices before and after the Iraq War were distinctly different. After the Iraq War, the international crude oil markets faced more volatility, which stimulated different speculative and arbitrage behaviors to influence the causal relationships. This result verifies the finding of Fan and Xu (2011), who pointed out that the Iraq War had been detected as a structural change point.
In pages pages 16-17 of our revised manuscript, we have reworded the statements as follows:
By comparing the results of causality in the mean and quantile causality, our study reveals three important findings: (a) the causal relationships between crude oil spot and futures prices show a heterogeneous behavior. Before the Iraq War, crude oil spot and futures prices were mutually Granger-caused at the lower quantile levels, indicating that both spot and futures prices provided useful information for forecasting the prices. However, at the upper quantile levels, the futures prices drove spot prices to the equilibrium level but not vice versa. After the Iraq War, in the higher quantile regions, the results reveal a clear bi-directional causality, i.e., both spot and futures prices react simultaneously to new information. However, when the oil markets are in the worst situation, only causality from futures to spot crude oil prices exists but not vice versa. The heterogeneous behaviors can provide more useful information to investors for hedging and investing in different market situations; (b) the causal relationships between the crude oil spot and futures prices present an obvious tail effect, i.e., either for the causal relationships from crude futures prices to spot prices or for reverse causality. The causality exists only over the lower and upper quantile intervals but not the middle quantile levels, impliying causality between crude oil spot and futures prices only for a high or low level of oil prices. Therefore, we can say that when the oil prices fluctuate near their medians, the causal relationships between spot and futures crude oil prices are weaker; (c) the causal relationships between crude oil spot and futures prices before and after the Iraq War are distinctly different. Regarding the causal relationships from crude oil futures prices to spot prices, before the Iraq War, there was significant causality over the lower and upper quantile intervals for all mature futures contracts. However, after the war, the causality existed only at the lower quantile levels for short-term contracts and at the upper quantile levels for longer-term contracts. Regarding the causal relationships from spot to futures crude oil prices, there is causality only at the lower quantile levels for short-term contracts before the war and only at upper quantile levels for longer-term contracts after the war. The different causalities before and after the Iraq War may be explained by the fact that the international crude oil markets faced more volatility after the war and stimulated different speculative and arbitrage behaviors.

Reviewer 3 Report
In this paper, the authors study the heterogeneous causal relationships between the crude oil spot and future crude oil prices in different market conditions. They state those causal relationships may differ depending on market conditions. Since investors usually are more sensitive to losses than to gains (Kahneman and Tversky,1979) the causality relationships between spot and futures crude oil prices may present heterogeneous effect across quantiles of crude oil futures and spot prices distribution. That is the reason the authors decide to study to what extent the heterogenity exists in the causalty relationships under a quantile causalty test approach. With this purpose, the authors apply a two stage approach; first they use a vector error correction model (VECM) to filter the long run comovement in the level of prices and then they examine the series of residuals by the quantile causality test proposed by Chuang et al. (2009). They test the causality relationships between crude oil spot and futures prices for before and after the onset of the Iraq war and show how this causal relationship change according to the onset of the Iraq war.They use weekly spot and futures prices for maturities of one , two, three and four months of West Texas Intermediate (WTI) which is used as benchmark in oil pricing and as the underlying commodity of New York Mercantile Exchange (NYMEX) for oil future contracts. Thus, using a two-stage quantile causality test method applied to crude oil spot and futures prices for the period March 1, 1986 to May 26, 2017, they find heterogeneous evidence regarding causal relationships between crude oil spot and futures prices. Results from the quantile causality test show that the significant causal relationships between crude oil spot and futures prices derive from tail quantile intervals. Moreover, they divide the sample period into two separate periods: Pre the Iraq war and Post the Iraq war sub-periods, they find there are causalities from futures prices to spot price in the lower and upper quantile levels for both before and after the Iraq war. However, regarding causality from spot price to futures prices, it is significant only in lower quantile levels before the Iraq war and only in upper quantile levels after the Iraq war. The topic is very interesting, the paper is well structured and written. However there are some minor spelling and grammar mistakes e.g. in line 461 "form" should be "from"
Author Response
Detailed Response to Reviewer 3
Dear Reviewer,
Thank you very much for reviewing the revised manuscript. We believe that your comments and suggestions are highly constructive and very helpful for our research. Thanks very much again for your detailed and thorough review. We have thoughtfully taken into your comments and responded to your constructive suggestions from point by point outlined below. We hope we have addressed all of your concerns.
For the sake of presentation, the comments of the referee are numbered and duplicated in italics, and our responses are given in plain. The page and line numbers of revised texts in our responses refer to our revised manuscript.
1. However there are some minor spelling and grammar mistakes e.g. in line 461 "form" should be "from".
Responses
Thank you for the detailed comments. We feel sorry for our spelling and grammatical errors in our earlier manuscript. Following you suggestion, we have carefully checked the spelling errors and carried out corresponding revision in our revised manuscript. In addition, we have used the English language editing service of MogoEdit to polish our paper.

Reviewer 4 Report
1. The contribution of this paper is marginal. I suggest that authors should emphasize the differences of proposed model between the paper and previous researches.
2. I suggest the author provide clear figure to describe the relationship between spot and futures oil prices. For example, figure 1 and figure 2 are not clear.
3. This paper applies two methods, the VECM model and quantile causality test. I suggest that the authors describe the methods more deeply. How to apply the methods to your data?
4. Please emphasize the differences between your findings and previous studies. Further, why this paper apply oil future market? Are the findings also consistent in other future markets?
Author Response
Detailed Response to Reviewer 4
Dear Reviewer,
Thank you very much for reviewing the revised manuscript. We believe that your comments and suggestions are highly constructive and very helpful for our research. Thanks very much again for your detailed and thorough review. We have thoughtfully taken into your comments and responded to your constructive suggestions from point by point outlined below. We hope we have addressed all of your concerns.
For the sake of presentation, the comments of the referee are numbered and duplicated in italics, and our responses are given in plain. The page and line numbers of revised texts in our responses refer to our revised manuscript.
1. The contribution of this paper is marginal. I suggest that authors should emphasize the differences of proposed model between the paper and previous researches.
Responses
Thank you very much for these constructive and insightful comments. In this study, we applied a two-stage approach to characterize the causal relationships between crude oil spot and futures prices. In the first step, we used a vector error correction model (VECM) to filter the long-run comovements in the level of prices. In the second step, we used the quantile causality test to examine the residuals of VECM. However, we did not emphasize the differences of proposed model between our paper and previous researches. In this revision, we emphasized the feasibility of our model and its differences with the previous method.
We hope we have addressed your concerns and the revision could fulfill your requirements. Thanks again for your comments.
In pages 3-4 of our revised manuscript, we have reworded the statements as follows:
To capture the heterogeneous causal relationships across different market conditions, we applied a two-stage approach to characterize the causal relationships between crude oil spot and futures prices. In the first step, we used a vector error correction model (VECM) to filter the long-run comovements in the level of prices. This is a reasonable setting for several reasons. On the one hand, it is quite well known that the spot and futures prices are cointegrated, so it is necessary to control the long-run comovement effects because they affect the specifications of the model used for causality testing. If the long-run comovement is not modeled, the evidence may vary significantly toward detecting linear and nonlinear causalities between the predictor variables (Bekiros and Diks, 2008). On the other hand, the VECM has been widely used in empirical studies to filter price series from long-run comovements (Hammoudeh et al., 2004; Bekiros and Diks, 2008; Kawamoto and Hamori, 2011; Serra et al., 2010; Hassouneh et al., 2012; Bekiros, 2014; Cabrera and Schulz, 2016). In the second step, after having filtered the price series with the VECM model, the series of residuals were examined with the quantile causality test proposed by Chuang et al. (2009). In contrast to a traditional linear estimation conditional on the mean distributions of dependent variables, this test allows us to explore the causality relationships conditional on the quantiles in the distributions of crude oil spot and futures prices. Because the quantile causality test considers different locations and scales of the conditional distribution, it may provide a complete description of the true causal relationships and can accommodate heterogeneous effects to allow us to capture the different responses of one market to another market at various quantile levels. Furthermore, the quantile levels for the conditional distribution of economic variables can indicate the states of the market and the quantile causality test provides an efficient tool for handling this nonlinearity in the causal relationships.
Our proposed two-stage causal analysis framework is one of the contributions of this paper to the existing literature. The previous models that took the market’s effects on the crude oil spot-futures relationships into account include mainly the structural break (Maslyuk and Smyth, 2009; Arouri et al., 2012; Chen et al., 2014), multivariate threshold regression (Huang et al., 2009; Wang and Wu, 2013), and quantile cointegrating models (Lee and Zeng, 2011). However, although the structural breaks and threshold methods can be used to explore the crude oil spot-futures relationships among special market states, how the causal relationships between spot and futures prices change across the quantiles of price distribution is little known. The quantile cointegration method allows for quantile-varying in a cointegrating vector and can offer much information for an understanding of the nonlinear relationships in various market conditions. However, the quantile cointegrating regressions can estimate the long-run relations between spot and future oil prices conditional on market innovations but cannot explain the short-term dynamic relationships under various market conditions. Therefore, we employed a VECM together with a quantile causality test to analyze the causal relationships between spot and futures prices on the basis of the performances of the crude oil markets. The two-stage model provides a flexible procedure and expands the available information set by considering long-run comovements and capturing various market states.
2. I suggest the author provide clear figure to describe the relationship between spot and futures oil prices. For example, figure 1 and figure 2 are not clear.
Responses
Thank you very much for your valuable comments. In our previous manuscript, we plot the crude oil futures prices and the differentials between spot and future prices in a same picture, which cannot clearly describe the relationships between spot and futures prices. Following your suggestion, we have redrawn fig. 2 to well describe the relationship between crude oil spot and futures prices. We hope we have addressed your concerns and the revision could fulfill your requirements.
In page 9 of our revised manuscript, we have redrawn the figures as follows:
WTI futures prices for Contract1 The differentials between spot and Contract1 prices
WTI futures prices for Contract2 The differentials between spot and Contract2 prices
WTI futures prices for Contract3 The differentials between spot and Contract3 prices
WTI futures prices for Contract4 The differentials between spot and Contract4 prices
Fig. 2. WTI futures prices and the differentials between spot and futures prices.
3. This paper applies two methods, the VECM model and quantile causality test. I suggest that the authors describe the methods more deeply. How to apply the methods to your data?
Responses
Thank you very much for your constructive comments. In our study, weapplied a two-stage approach to characterize the causal relationships between crude oil spot and futures prices. First, we used a vector error correction model (VECM) to filter the long-run comovements in the level of prices. Second, we used quantile causality test to examine the causality between residuals of VECM. In order to well describe how to apply the methods to our data, we more deeply describe our methods in this revised manuscript. We hope we have addressed your concerns and the revision could fulfill your requirements.
In pages 6-7 of our revised manuscript, we have reworded the statements as follows:
The concept of cointegration has addressed the characteristic comovement of commodity price series. Cointegration is the idea that, even though individual price series are non-stationary, a linear combination of price series may be stationary. The VECM is a VAR model with cointegration restrictions. Moreover, the VECM includes an examination of the dynamic comovements among variables and the adjustment process toward long-term equilibrium. To filter the crude oil spot and futures prices from long-run comovements, we applied a VECM to specify the conditional mean.
Let be a vector of price series, denotes the crude oil spot price at time , denotes the crude oil futures price at time . If the elements of are variables, then the conventional VECM is represented as
(1)
After having filtered the price series with the VECM model, the series of VECM residuals were examined with the quantile causality test. In the following we will assume that and is the VECM residuals. For a comprehensive understanding of causal relationship between and , the causality test in quantile proposed by Chuang et al. (2009) is represented as
(3)
4. Please emphasize the differences between your findings and previous studies. Further, why this paper apply oil future market? Are the findings also consistent in other future markets?
Responses
Thank you very much for these constructive and insightful comments. There are a lot of literatures studying the causal relationship between crude oil spot and futures prices in the past several decades. The biggest difference between our study and previous studies is that we mainly consider the causal heterogeneity caused by different market conditions. To address this task, we use a two-step quantile causality analysis framework, in which the VECM be used to filter the long-run comovement. In pages 4-5 of our revised manuscript, we emphasize our findings and its differences with previous studies.
The reason why this paper applies oil futures market is that the causal relationships between crude oil spot and futures prices may differ according to market conditions. In the crude oil spot-futures markets, if the relationships between the spot and futures prices do not approach equilibrium, then effects such as arbitrage and speculation will produce equilibrium. Therefore, the causality relationships between spot and futures crude oil prices may present heterogeneous effects across the quantiles of crude oil spot and futures price distributions. This possibility inspired us to apply a quantile causality analytical framework to investigate the extent to which heterogeneity exists in the causality relationships. Moreover, we believe that the finding that different market conditions can lead to heterogeneous causal relationships should also be true for other futures markets. In future studies, we will study the heterogeneous effects of different market conditions on other futures markets.
We hope we have addressed your concerns and the revision could fulfill your requirements. Thanks again for your comments.
In pages 4-5 of our revised manuscript, we have reworded the statements as follows:
The key findings of our paper are as follows. First, we provide new evidence that the causal relationships between crude oil spot and futures prices present significant heterogeneous behaviors under different market conditions. We have observed that the causal relationships between crude oil spot and futures prices over only the lower and upper quantile intervals but not the middle quantile intervals. The causal relationships between crude oil spot and futures prices are weaker when oil prices fluctuate at nearly their medians, and are highly likely to be present when one market shows very good or very poor performance. This empirical result expands on and refines the findings of Huang et al. (2009), as well as those of Wang and Wu (2013), who suggested that at least one causal relationship between spot and futures prices exists only when the price differentials are larger than a threshold value. Our results were able to capture the causal relationships between crude oil spot and futures prices at specified quantile levels. Moreover, as compared to the conclusion of Lee and Zeng (2011), we found that crude oil spot and futures prices were influenced by quantiles and futures contracts, but they had found that spot oil prices indeed caused futures oil prices. This finding was not apparent in our empirical results. The difference in the sample interval we had selected may be the underlying reason for the inconsistency.
Second, we have revealed the nonlinear causality between crude oil spot and futures prices from the perspectives of different market states. We found that the direction and significance of causal relationships between crude oil spot and futures prices changed with different market conditions. This result is in accord with Bekiros and Diks (2008), who suggested that if nonlinear effects were accounted for, neither market led or lagged behind the other consistently. Chang and Lee (2015) used a wavelet coherency method and studied the time-varying causal relationships between crude oil spot and futures prices. Their results show significant dynamic causality between variables in the time-frequency domain. Here, we consider the nonlinear drivers, which are different from the focus of previous studies, and obtain a consistent conclusion.
Third, we observed that the causal relationships between crude oil spot and futures prices before and after the Iraq War were distinctly different. After the Iraq War, the international crude oil markets faced more volatility, which stimulated different speculative and arbitrage behaviors to influence the causal relationships. This result verifies the finding of Fan and Xu (2011), who pointed out that the Iraq War had been detected as a structural change point.

Round 2
Reviewer 4 Report
Journal: Sustainability-418537 (2nd round)
Title: Heterogeneous causal relationships between spot and futures oil prices: Evidence from Quantile causality analysis
Manuscript Number: sustainability-418537
The study is quite interesting, and the outcomes of this paper are a valuable addition to literature. I have some comments as follows:
1. The authors should avoid using personal pronoun ‘we.’ However, the authors used ‘we’ in several places in this paper. For example, the authors said, “we investigated.’ It prefers to say this paper/study investigated. That is, the authors used personal pronoun ‘we’ or ‘our’ throughout the paper and should replace with the word “this study or this research”
2. What are the populations of your data (Sampling frame, total population, etc.)? Why specifically was this sample period selected for this study? Why have crude oil been chosen?
3. I cannot see the horizontal axis of figure 1 and figure 2 in revised manuscript.

Author Response
Detailed Response to Reviewer 4 (2nd round)
Dear Reviewer,
Thank you very much for reviewing the revised manuscript. We believe that your comments and suggestions are highly constructive and very helpful for our research. Thanks very much again for your detailed and thorough review. We have thoughtfully taken into your comments and responded to your constructive suggestions from point by point outlined below. We hope we have addressed all of your concerns.
For the sake of presentation, the comments of the referee are numbered and duplicated in italics, and our responses are given in plain. The page and line numbers of revised texts in our responses refer to our revised manuscript.
1. The authors should avoid using personal pronoun ‘we.’ However, the authors used‘we’ in several places in this paper. For example, the authors said, “we investigated.’ It prefers to say this paper/study investigated. That is, the authors used personal pronoun ‘we’ or ‘our’ throughout the paper and should replace with the word “this study or this research”.
Responses
Thank you very much for your valuable comments. We have replaced the personal pronoun ‘we’ with the words ‘this study’ or ‘this research’ in the 2nd round revision. We hope we have addressed your concerns and the revision could fulfill your requirements. Thanks again for your comments.
2. What are the populations of your data (Sampling frame, total population, etc.)? Why specifically was this sample period selected for this study? Why have crude oil been chosen?
Responses
Thank you very much for these constructive and insightful comments. Crude oil is a major energy source in the world today. It is important to further understand the price discovery mechanism in crude oil spot and futures markets. However, the available empirical evidence on the causal relationships between oil spot and futures prices is mixed. Therefore, we chose crude oil as our research object. In this study, our main purpose is to examine the heterogeneous causal relationships between crude oil spot and futures prices under different market conditions.
The populations of our data should be the spot and futures prices of international crude oil. In this study, we used the WTI (West Texas Intermediate) spot and futures prices covering period from March 1, 1986 to May 26, 2017 as a sample. WTI is a type of crude oil used as the benchmark in pricing and is the underlying commodity of the oil futures contracts at the New York Mercantile Exchange’s (NYMEX), whose market provides important price information to buyers and sellers of crude oil around the world. WTI spot and futures prices have been used by many researchers as sample to study causality between crude oil spot and futures prices (see e.g., Huang et al., 2009; Wang and Wu, 2013; Lee and Zeng, 2011; Chang and Lee 2015).
In this study, the samples cover the period from March 1, 1986 to May 26, 2017. The starting date determined by data availability from the website of the U.S. Energy Information Administration (EIA). In addition, this sample period includes March 21, 2003, when the Iraq War broke out. After the Iraq War broke out, the average level and volatility of WTI prices showed an accelerated rise. The selection of sample period provides sufficient samples for us to analyze the relationship between crude oil spot and futures prices before and after the Iraq War.
References:
Huang B N, Yang C W, Hwang M J. 2009. The dynamics of a nonlinear relationship between crude oil spot and futures prices: A multivariate threshold regression approach. Energy Economics 31(1): 91-98.
Wang Y, Wu C. 2013. Are crude oil spot and futures prices cointegrated? Not always! Economic Modelling 33: 641-650.
Lee C C, Zeng J H. 2011. Revisiting the relationship between spot and futures oil prices: evidence from quantile cointegrating regression. Energy economics 33(5): 924-935.
Chang C P, Lee C C. 2015. Do oil spot and futures prices move together? Energy economics 50: 379-390.
3. I cannot see the horizontal axis of figure 1 and figure 2 in revised manuscript.
Responses
Thank you very much for your valuable comments. In our previous revision, we didn’t show clearly the names of the abscissa and ordinate in Fig. 1 and Fig.2. In the 2nd round revision, we have redrawn Fig.1 and Fig.2 to well describe the relationship between crude oil spot and futures prices. We hope we have addressed your concerns and the revision could fulfill your requirements.
In page 8-9 of our revised manuscript, we have redrawn the fig.1 and fig.2 as follows:
Fig. 1. WTI crude oil spot prices.
WTI futures prices for Contract1 The differentials between spot and Contract1 prices
WTI futures prices for Contract2 The differentials between spot and Contract2 prices
WTI futures prices for Contract3 The differentials between spot and Contract3 prices
WTI futures prices for Contract4 The differentials between spot and Contract4 prices
Fig. 2. WTI futures prices and the differentials between spot and futures prices.
